# Two New Datasets for Italian-Language Abstractive Text Summarization

Nicola Landro [1,2,†,‡] , Ignazio Gallo [1,*,†,‡] , Riccardo La Grassa [3,†] and Edoardo Federici [4,†]

1 Department of Theoretical and Applied Sciences—DISTA, University of Insubria, Via J.H. Dunant, 3, 21100 Varese, Italy; nlandro@uninsubria.it
2 AIKnowYOU srl, 21029 Varese, Italy
3 INAF-Osservatorio Astronomico di Padova, Vicolo dell'Osservatorio, 35122 Padova, Italy; rlagrassa@uninsubria.it
4 Digitiamo srl, 21100 Varese, Italy; efederici1@studenti.uninsubria.it
* Correspondence: ignazio.gallo@uninsubria.it
† These authors contributed equally to this work.
‡ Current address: Dipartimento di Scienze Teoriche e Applicate—DISTA, University of Insubria, Via O. Rossi, 9—Padiglione Rossi, 21100 Varese, Italy.

**Abstract:** Text summarization aims to produce a short summary containing relevant parts from a given text. Due to the lack of data for abstractive summarization on low-resource languages such as Italian, we propose two new original datasets collected from two Italian news websites with multi-sentence summaries and corresponding articles, and from a dataset obtained by machine translation of a Spanish summarization dataset. These two datasets are currently the only two available in Italian for this task. To evaluate the quality of these two datasets, we used them to train a T5-base model and an mBART model, obtaining good results with both. To better evaluate the results obtained, we also compared the same models trained on automatically translated datasets, and the resulting summaries in the same training language, with the automatically translated summaries, which demonstrated the superiority of the models obtained from the proposed datasets.

**Keywords:** abstractive text summarization; datasets; deep learning

## 1. Introduction

Text summarization aims to produce a summary from an existing text. With a model that summarizes text, we can save time and obtain more useful insights from data. We would also be able to store less information and could make better use of the information we store. In text mining, accurate and coherent summaries are very useful for information extraction. Abstractive summarization [1] and extractive summarization [2] are two different approaches to the same task. On one hand, summarization is called *abstractive* when the summary generated is not a selection of sentences from the input document, but a more cohesive paraphrase containing compressed information, using also a partially different vocabulary; on the other hand, summarization is called *extractive* when we compute a score for input sentences and produce a summary that concatenates sentences that have obtained higher scores on some metric. Recent papers in abstractive summarization mainly focus on English text, due to the abundance of English-language datasets (such as XSum [3] and CNN/DM [4]) and pre-trained models. In order to better understand how abstractive summarization techniques work, regardless of the language used, it is important to have well-made datasets, even in languages other than English. For this reason, we propose two datasets in Italian, and make them available to the scientific community.

In the past few years, sequence-to-sequence models obtained state-of-the-art results on multiple language tasks, such as translation, text-generation, and summarization [5]. Recent trends in machine learning for text mining go in the direction of using transformer-based models [6], which are increasingly larger in terms of the number of parameters (see, for

example, models such as T5 [7], Turing NLG [8], GPT-3 [9], Megatron-Turing NLG [10], and so on). This trend makes the training process very difficult because it requires computing power that only a few can afford. A solution to this problem is the fine-tuning of pre-trained models on tasks similar to our problem. Unfortunately, big transformer-based [10] models are mostly pre-trained in English, resulting in them being ineffective for summarization of texts in low-resource languages, such as the Italian language. This leads us to say that having datasets in Italian is also a solution to this problem, giving us the possibility of releasing transformer-based models that are pre-trained on the Italian language, and that are easy to use by others, by exploiting the fine-tuning.

In this paper, we have used the IT5 model [11], a generic T5-base model [7] for the Italian language that is not explicitly trained for summarization. IT5 is publicly available on HuggingFace, trained on the Italian part of the corpus mC4 [12]. We also use mBART [13], which is already trained on multiple languages. We have fine-tuned these two models on the datasets we propose here, obtaining good results that are almost comparable to the state-of-the-art results of the same models fine-tuned in English.

## 2. Related Work

### 2.1. Models

In the absence of fine-tuned models for abstractive summarizations in Italian, extractive methods were the only possible approach. Extractive methods produce summaries by the concatenation of meaningful sentences from an analyzed text. Therefore, the task is reduced to computing metrics on various text sentences and selecting the best sentence based on the score they obtained. TextRank [14], as an example, is an unsupervised graph-based content extraction algorithm. A graph is constructed where nodes of the graph represent each sentence and edges represent content-overlap (obtained by computing the number of in-common words). Starting from this network, sentences are fed into a PageRank [15] algorithm, which identifies the most relevant ones. This approach is useful also for texts with a high number of sentences, as it computes Jaccard distances and applies the Minhash algorithm to obtain a smaller number of sentence combinations.

The approach mentioned above is a quite different task than the abstractive summarization we are interested in. For abstractive summarization, sequence-to-sequence pre-trained models based on a transformer architecture have shown great success. Recent techniques, using T5 [7], Pegasus [16], and ProphetNet [17], showed the best results. T5 (Text-to-Text Transfer Transformer) is trained for text-to-text problems. It applies a unified model and a training procedure to a variety of NLP tasks, such as generating similar sentences, completing a story, etc. [18]. It was pre-trained using denoising objectives (masking a sequence of words from the sentence and training the model to predict the masked words) that showed the most promising results. Pegasus was inspired by the T5 approach. In Pegasus, the pre-training objective is replaced by creating a pseudo-summary after masking whole important sentences from a text and concatenating them as summaries (gap-sentences).

BART [19] is another possible choice for abstractive summarization. In fact, BART is a deep model that can be trained on a generic task and, therefore, can be fine-tuned for different tasks. In the original paper, mBART [13] was used for multilingual machine translation. Since, among the many languages, mBART has also been trained to translate into Italian, it is possible to fine-tune it for text summarization in Italian, knowing that BART was used for text summarization.

In conclusion, IT5 and mBART are good for summarizing tasks, and they occupy relatively few resources because there is a pre-trained model in Italian and we have selected them to run tests on the datasets we propose. In addition, we also use results from Pegasus for comparison with our results. Pegasus is a very large model, there is no pre-trained model for it in Italian, and it is too expensive for us to train it from scratch; thus, we use it for translations made by the OPUS-MT [20] models: the input text was translated into English (It–Eng Translator: https://huggingface.co/Helsinki-NLP/opus-mt-it-en, ac-

cessed on 23 March 2022), and the output text was translated into Italian (Eng–It Translator: https://huggingface.co/Helsinki-NLP/opus-mt-en-it, accessed on 23 March 2022).

*2.2. Datasets Used*

In recent years, many datasets have been created for text summarization for the English language. In the previous sections, we have already mentioned XSum [3] and CNN/DM [4]. The XSum dataset is intended for evaluating the abstractive summarization systems of single documents. In this dataset, the goal is to create a new, short, one-sentence summary that answers the question "What is this article about?". The dataset consists of 226,711 news articles, each accompanied by a one sentence summary. The articles were sourced from the BBC (2010 to 2017), and cover a wide variety of domains (e.g., news, politics, sports, weather, business, technology, science, health, family, education, entertainment, and art). CNN/Daily Mail is a dataset for text summarization. The human-generated abstractive summary was generated from news on the CNN and Daily Mail websites, such as questions and stories. In all, the corpus has 286,817 training pairs, 13,368 validation pairs, and 11,487 test pairs. Source documents in the training set have 766 words, covering an average of 29.74 sentences, while summaries consist of 53 words and 3.72 sentences.

There are also multilingual datasets for text summarization, such as MLSum [21], obtained from online newspapers and containing 1.5 million article/summary pairs in five different languages: French, German, Spanish, Russian, and Turkish. Italian is omitted in all available multilingual datasets, and the main reason for this depends on the source from which these datasets are extracted. In fact, most of these datasets use sites, such as the BBC's website, to create document/summary pairs in the various languages; the BBC's site does not report the news in Italian. To compensate for this lack, we have decided to create and make public datasets in Italian that are useful for the scientific community.

## 3. Models and Metrics

A text-to-text approach uses the same model, the same loss function, and the same hyperparameters across all NLP activities. In this approach, the inputs are modeled in such a way that the model recognizes a task and the output is simply the "textual" version of the expected result. So, in a text-to-text approach, the same model (for example, T5) can be used to make translations from one language to another, or to conduct text summarization, or for any other activity that transforms textual information into other text. Problems such as summarization, translation, and question-answering can be modelled as text-to-text problems. We selected a T5 (**T**ext-**t**o-**T**ext **T**ransfer **T**ransformer) base model (IT5) pre-trained on the Italian portion of mC4 [7], which is a very large dataset consisting of natural text documents in 101 languages, and is also a variant of the "Colossal Clean Crawled Corpus" (C4), which is a dataset consisting of hundreds of gigabytes of clean English text scraped from the web. IT5 is first pre-trained on the mC4 dataset for the denoising and corrupting span objective with an encoder–decoder architecture. It is then fine-tuned on the downstream tasks with a supervised objective with appropriate input modeling for the text-to-text setting. In particular, we fine-tuned IT5 for abstract summarizations on all of our datasets. IT5 is the first model pre-trained on mC4, and we found it suitable for our task.

As the second model, we used mBART, which is a sequence-to-sequence denoising auto-encoder pre-trained once for all languages (including Italian), providing a set of parameters that can be fine-tuned for any of the language pairs. In this case, we have exploited the Transformers library [22], which makes it easy to use these models.

The summaries produced with the identified models must be evaluated and compared with the results of other models. The ROUGE [23] metrics are commonly used in this context. The ROUGE (Recall-Oriented Understudy for Gisting Evaluation) [23] metrics are for evaluating summaries; they have two typical forms, ROUGE-N and ROUGE-L.

ROUGE-N can be seen as an n-gram recall between a candidate summary and a set of reference summaries. It can be defined as Equation (1):

$$\text{ROUGE-N} = \frac{\sum_S^{\text{Ref. Sum.s}} \sum_{gram_n}^S Count_{match}(gram_n)}{\sum_S^{\text{Ref. Sum.s}} \sum_{gram_n}^S Count(gram_n)} \tag{1}$$

where $n$ is the length of the $gram_n$ (the numbers of words in each gram), $Count$ is the function that counts the occurrence of an n-gram into the summary, and $Count_{match}$ is the maximum number of n-grams co-occurring in a candidate summary and a set of reference summaries. ROUGE-L counts the longest common sub-sequences (LCS). Given two sequences, $X$ and $Y$, the LCS is a common sub-sequence with the maximum length. To port this concept to summaries, we use a LCS-based F-measure that estimates the similarity between two summaries $X$ of length $m$ and $Y$ of length $n$, assuming $X$ is the reference and $Y$ is the candidate summary:

$$R_{lcs} = \frac{LCS(X,y)}{m} \tag{2}$$

$$P_{lcs} = \frac{LCS(X,y)}{n} \tag{3}$$

$$F_{lcs} = \frac{(1+\beta^2)R_{lcs}P_{lcs}}{R_{lcs} + \beta^2 P_{lcs}} \tag{4}$$

where $LCS(X,Y)$ is the length of the longest common sub-sequence of $X$ and $y$; the $\beta$ is set to $\frac{P_{lcs}}{R_{lcs}}$ if $\frac{F_{lcs}}{R_{lcs}} = \frac{F_{lcs}}{P_{lcs}}$. Otherwise, only $R_{lcs}$ is considered. The LCS-based F-mesure in Equation (4) is called ROUGE-L. This metric is computed between two summaries, so if we have a reference and multiple candidates, the average is normally used convert it to a number. For this reason, we also report a version of ROUGE-L that uses the sum instead of the mean, and we call it ROUGE-LS. In our experiments, we also calculate the average generation length because, in deep models, the generation length is generally less than a predetermined maximum length, so it is important to understand how many words these models use in a summary.

Our code is published on GitLab [24], and our trained models are published on HuggingFace [25–30]. A demo of trained models is available online (Demo: https://huggingface.co/spaces/ARTeLab/ARTeLab-SummIT, accessed on 23 March 2022).

## 4. Proposed Datasets

In this section, we present two new original corpora for Italian abstractive summarization. Large-scale summarization datasets are rare, and for low-resource languages such as Italian, prior to this paper, there were none available. We collected data via web-scraping on two of the major Italian news sites, whose articles are freely available.

We propose two new datasets Fanpage (https://huggingface.co/datasets/ARTeLab/fanpage) and IlPost (https://huggingface.co/datasets/ARTeLab/ilpost) with multi-sentence summaries, i.e., datasets for text summarization in which the summary created as truth can contain more than one sentence. These two datasets are extracted from Fanpage (https://www.fanpage.it/) and IlPost (https://www.ilpost.it/), respectively. The first has an average summary length of 1.96 sentences and 43.85 words, while the second dataset has an average summary length of 1.91 sentences and 26.39 words (see Table 1 for other statistics) (all accessed on 23 March 2022).

Inspired by [3,31], the methodology adopted to create a large-scale supervised dataset for text summarization is very simple because it uses a fixed layout of a single website instead of scraping documents from the web. A key point of the datasets we have created is that the summaries contained in them must be abstractive and, therefore, must not be extractive, i.e., a subset of sentences extracted from the input text. Therefore, during the creation, we used the ROUGE-L metrics to check if the sentences of the summary could be

found within the text of the article, discarding all the article/summary pairs with a value beyond the fixed ROUGE-L threshold.

**Table 1.** Average number of words and average length in terms of number of phrases and vocabulary size for the MLSum-It, IlPost, and Fanpage datasets.

|  | Dataset | Input Text | Summary |
|---|---|---|---|
| Avg. num. of words | MLSum-It | 184.73 | 17.05 |
|  | IlPost | 174.43 | 26.39 |
|  | FanPage | 312.70 | 43.85 |
| Avg. num. of sentences | MLSum-It | 6.07 | 1.06 |
|  | IlPost | 5.88 | 1.91 |
|  | FanPage | 11.67 | 1.96 |
| Vocabulary size | MLSum-It | 308,419 | 67,507 |
|  | IlPost | 309,609 | 97,099 |
|  | FanPage | 730,850 | 198,649 |

In order to use the datasets, we took some simple preprocessing steps, and then generated the dictionaries for each dataset (see details in Table 1) after having divided each dataset into 80% for training, 10% for validation, and 10% for the test set. Each document in the dataset was first converted to lowercase, and then all punctuation characters were removed.

*4.1. IlPost*

On the IlPost website, all articles are structured in the same way: they have a title, a short description, and a complete description of the news. As reported in Table 1, the average number of sentences for the news of this website is equal to 5.88 sentences per news item, while the short description contains, on average, 1.91 sentences per news item. The same Table 1 also reports the average number of words for the short description (summary) and for the news (input document). To create the summary of the news, we concatenated the title and the short description available on this website, while we used the text of the news as the source document. Figure 1 shows an example of a title and short description extracted from the IlPost website. As reported in Table 2, the dataset contains 7 different categories of news, each of which has a variable number of documents for a total of 44,025 documents and corresponding summaries, which makes it a very interesting and useful dataset for training a deep neural model.

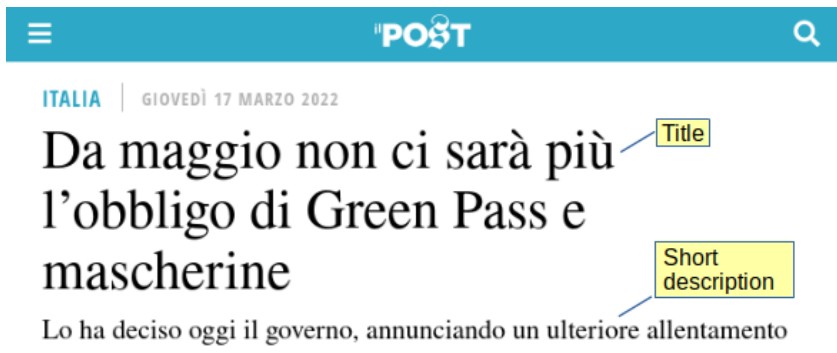

**Figure 1.** An example of an article downloaded from the IlPost news website. The figure highlights the title and the short description, which are usually followed by an image and the full article.

The text of each document and each summary available in the dataset has been pre-processed with the text preprocessing library Textacy (https://textacy.readthedocs.io/en/latest/, accessed on 23 March 2022) to clean up any HTML code left in the documents.

**Table 2.** Document cardinalities of the IlPost and Fanpage datasets, divided into different categories. The total cardinality is also reported for the two datasets and for the translated MLSum-It dataset.

| | Number of Documents | | |
|---|---|---|---|
| **Topic** | **IlPost** | **FanPage** | **MLSum-It** |
| Tech (Tech) | 2497 | 3776 | - |
| Scienza (Science) | 3849 | 5157 | - |
| Italia (Italy) | 10,739 | - | - |
| Politica (Politics) | 4759 | - | - |
| Internet (Web) | 2939 | - | - |
| Economia (Economy) | 3823 | - | - |
| Cultura (Culture) | 15,419 | - | - |
| Misc (Miscellaneous) | - | 23,150 | - |
| Travel (Travel) | - | 1535 | - |
| Donna (Whoman) | - | 9104 | - |
| Design (Design) | - | 5946 | - |
| Musica (Music) | - | 12,034 | - |
| Gossip (Gossip) | - | 15,770 | - |
| Cinema (Cinema) | - | 7936 | - |
| Total | 44,025 | 84,308 | 40,000 |

*4.2. Fanpage*

For the Fanpage site, each news item is structured in the same way from the point of view of the document structure, that is, taking into consideration the title, the short summary, and a complete description of the news. Unlike the URL, where the articles belonging to the same topic (see list of topics in Table 2) have the same base URL address, the URL address changes from one topic to another (e.g.,: for "Tech", we can find articles on www.tech.fanpage.it, accessed on 23 March 2022). For this reason, the built crawler required more attention in the download phase, so we scrapped all the text–abstract pairs from the HTML source. Since the summaries were easily identifiable with *"fp_intro__abstract"* within the HTML of each article, we had no particular difficulty with obtaining this dataset. As reported in Table 1, the average number of sentences for the news of this web site is equal to 11.67 phrases per news item (i.e., much longer than the 5.88-phrase average of the IlPost dataset), while the shortest summaries contain an average of 1.96 sentences each. The same Table 1 also reports the average number of words for the short summary and for the news. To create the summary of the news, we joined the title and the brief description available on this site, while we used the text of the news as the input document. Figure 2 shows an example title and short summary extracted from the Fanpage website. As reported in Table 2, the dataset contains 9 different categories of news (two in common with those of the IlPost dataset), each of which has a variable number of documents, for a total of 84,308 documents and related summaries (roughly double the amount in the IlPost dataset), which makes it a very interesting and useful dataset for training a deep neural model.

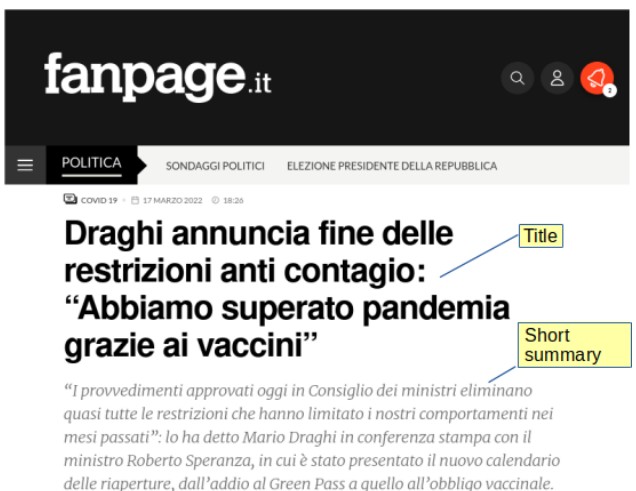

**Figure 2.** An example of an article downloaded from the Fanpage news website. The figure highlights the title and the short summary.

*4.3. MLSum-It*

The IlPost and Fanpage datasets, created ad hoc for text summarizations for the Italian language, raise the question of whether we really need to build a dataset specifically for Italian, or if we can use automatic translation models to create them automatically, starting from already-existing datasets that are available in another language. To answer this question, we have translated the MLSum dataset [21], a famous multilingual dataset for large-scale text summarization, creating MLSum-It (https://huggingface.co/datasets/ARTeLab/mlsum-it), an Italian-language version, starting from Spanish, which is a language very similar to Italian. The original MLSum was mainly obtained from articles downloaded from BBC/Mundo. MLSum-It was obtained via machine translation using the OPUS-MT [20] model (Es-It Translator: https://huggingface.co/Helsinki-NLP/opus-mt-es-it, all accessed on 23 March 2022). As reported in Table 2, the total number of documents contained in MLSum-It is equal to 40,000.

**5. Experiments**

In this section, we want to conduct experiments to evaluate the quality of the proposed datasets. In particular, we want to answer the following questions:

- Is it better to translate the dataset to the target language or to the input/output for the pre-trained model trained with a different target language?
- Is it better to train a "simple" model on a dataset created in the original target language, or is it better to use a SOTA model trained with a different target language, and then automatically translate the input/output for it?
- Is it worth creating a dataset in the target language, or is it better to automatically translate a dataset from a different language into the target language?

Throughout all of the experiments conducted to answer these questions, we used the following configurations. We used PyTorch on a machine with three NVIDIA GPUs. The input documents were all truncated at 512 tokens. We used a maximum length of 128 tokens for the generated summaries. We used a learning rate equal to 0.00005, an Adam optimizer with $\beta$s = (0.9, 0.999) and $\epsilon = 1 \times 10^{-8}$, and a linear learning rate scheduler with a max number of epochs = 4. The batch size was set to six during training, except for nBART, where we used a batch size equal to one.

To answer the first question, we trained the two IT5 and mBERT models on the MLSum-It dataset, translated from Spanish into Italian, and then we compared the results with those produced by the two Pegasus models reported in Table 3 and described in Section 2. Table 3 reports the results in terms of various ROUGE measures and also analyzes the length of the summaries produced. From the numerical results reported in Table 3, we can

concluded that the IT5 and mBART models trained on MLSum outperform the Pegasus results. In particular, all the metrics are better, with the exception of the average length of the summaries produced by Pegasus, which are longer. In conclusion, we can say that training a model on a dataset, even if automatically translated from another language other than the target one, produces better results than those produced by a model trained on a dataset in a different language from the destination one, even if the model is very powerful.

**Table 3.** Comparison between IT5, mBART, Pegasus-XSum and Pegasus-CNN/DM using the same MLSum-It test set. IT5 and mBART are trained on MLSum-It, while the two Pegasus models are pre-trained. |Summary| is the length of the summary in terms of the number of words. The best results reported in the table are highlighted in bold.

| MLSum-It | IT5 | mBART | Pegasus-XSum | Pegasus-CNN/DM |
|----------|-----|-------|--------------|----------------|
| ROUGE-1 | 19.29 | **19.35** | 15.17 | 16.97 |
| ROUGE-2 | 6.04 | **6.40** | 3.57 | 5.03 |
| ROUGE-L | **16.50** | 16.35 | 12.45 | 13.11 |
| ROUGE-LS | **16.62** | 16.54 | 12.44 | 13.11 |
| |Summary| | 32.76 | 33.59 | 35.03 | **81.04** |

The Fanpage and IlPost datasets proposed in this paper allow us to answer the second question and, at the same time, allow us to better understand the quality of the proposed datasets. The results reported in Tables 4 and 5 show higher ROUGEs compared to the results in Table 3, so we can deduce that a dataset created on the target language is better than the translated one. Furthermore, the metrics of the models trained on the datasets in Italian remain much higher than those reported by Pegasus. One thing that we can notice better from the Tables 4 and 5 is that the length of the generated text is strictly dependent on the dataset (more than on the max length parameter of the model); in fact, on the IlPost dataset, we have a lower generation length for both mBART and IT5 than for the one we find on Fanpage.

**Table 4.** Comparison between IT5, mBART, Pegasus-XSum, and Pegasus-CNN/DM, using the same Fanpage test set. IT5 and mBART are trained on Fanpage, while the two Pegasus models are pre-trained. |Summary| is the length of the summary in terms of the number of words. The best results reported in the table are highlighted in bold.

| Fanpage | IT5 | mBART | Pegasus-XSum | Pegasus-CNN/DM |
|---------|-----|-------|--------------|----------------|
| ROUGE-1 | 33.83 | **36.50** | 20.01 | 26.82 |
| ROUGE-2 | 15.46 | **17.44** | 6.49 | 9.02 |
| ROUGE-L | 24.90 | **26.17** | 14.78 | 18.10 |
| ROUGE-LS | 28.31 | **30.26** | 14.76 | 18.10 |
| |summary| | 69.80 | 75.24 | 32.33 | **80.50** |

**Table 5.** Comparison between IT5, mBART, Pegasus-XSum, and Pegasus-CNN/DM, using the same IlPost test set. IT5 and mBART are trained on IlPost, while the two Pegasus models are pre-trained. |Summary| is the length of the summary in terms of the number of words. The best results reported in the table are highlighted in bold.

| IlPost | IT5 | mBART | Pegasus-XSum | Pegasus-CNN/DM |
|--------|-----|-------|--------------|----------------|
| ROUGE-1 | 33.78 | **38.91** | 21.03 | 23.96 |
| ROUGE-2 | 16.29 | **21.38** | 6.63 | 7.72 |
| ROUGE-L | 27.48 | **32.05** | 16.10 | 16.81 |
| ROUGE-LS | 30.23 | **35.07** | 16.07 | 16.81 |
| |summary| | 45.32 | 39.88 | 29.79 | **77.53** |

To answer the third and final question, and to better understand if a dataset created directly in the Italian language is more robust than a dataset translated into Italian, we performed cross-dataset experiments in Table 6. In particular, given the test set of a certain dataset, we have given it as an input to the models trained on the training sets of different datasets. From the results shown in Table 6, we can see that, on the IlPost test set, both IT5 and mBART trained on Fanpage exceed the results of the same models trained on MLSum-It. The same thing can be deduced from the results obtained on the Fanpage test set, i.e., the models trained on IlPost exceed the results of the same models trained on MlSum-it. While in the last part of Table 6, on the MLSum-It test set, we see very similar ROUGE values, which, in any case, differ slightly between the model trained on Fanpage and the one trained on IlPost (the difference in ROUGE-1 is approximately 0.6), so we can deduce that, apart from the average length generated, the two datasets are equivalent. In conclusion, we can say that it is always better to create a dataset directly for the target language. We can, therefore, see the two datasets proposed here as necessary datasets for all those who want to create text summarization models for the Italian language.

**Table 6.** Comparison of IT5 and mBART models when trained on a dataset and tested on another dataset. All combinations of the test and training sets of the MLSum-It, Fanpage, and IlPost datasets are reported in the table. The metrics reported are ROUGE-1 (R-1), ROUGE-2 (R-2), ROUGE-L (RL), ROUGE-LS (R-LS), and the average length of the summaries (|sum.|).

| Trainset | Model | R-1 | R-2 | R-L | R-LS | \|sum.\| |
|---|---|---|---|---|---|---|
| | | | IlPost test set | | | |
| Fanpage | IT5 | 23.62 | 10.91 | 19.65 | 19.65 | 19.0 |
| MLSum-It | IT5 | 19.58 | 7.56 | 16.53 | 16.53 | 18.98 |
| Fanpage | mBART | 29.36 | 12.12 | 21.01 | 21.01 | 75.9 |
| MLSum-It | mBART | 24.69 | 8.91 | 18.69 | 18.69 | 39.72 |
| | | | Fanpage test set | | | |
| Il Post | IT5 | 20.57 | 9.33 | 16.76 | 16.76 | 18.99 |
| MLSum-It | IT5 | 17.4 | 7.4 | 14.66 | 14.66 | 18.97 |
| Il Post | mBART | 29.33 | 11.3 | 20.46 | 20.46 | 45.09 |
| MLSum-It | mBART | 23.4 | 8.73 | 17.58 | 17.58 | 35.34 |
| | | | MLSum-It test set | | | |
| Fanpage | IT5 | 15.13 | 4.83 | 13.6 | 13.6 | 19.0 |
| Il Post | IT5 | 15.77 | 4.94 | 13.97 | 13.97 | 18.99 |
| Fanpage | mBART | 18.64 | 6.13 | 14.51 | 14.51 | 80.48 |
| Il Post | mBART | 19.24 | 5.52 | 15.38 | 15.38 | 46.26 |

*Human Evaluation*

In addition to the evaluation metrics used, we also show and visually analyze some summary examples produced by the various models used in this paper. To allow readers to deepen this visual and numerical analysis, we also make the code, the trained models, and a web page with a directly usable demo available to the scientific community.

As the first document to be analyzed, we took the fifth example from the Fanpage test set, and obtained the summary generated by the IT5 model trained on Fanpage. The produced summary is as follows:

> *"Mary Katherine, una giovane ragazza alle prese con il mondo sconosciuto della flora che ci circonda, si risveglia un giorno e scopre di non essere più a casa sua, ma di essere stata*

> *trasportata in un altro universo. un viaggio epico, con un cast di doppiatori originali da brivido."*

The summary produced in this case was defined by a human being as "a great summary with no grammatical errors", practically very similar to a summary produced by a human. We can compare the previous summary generated by the model with its expected summary or ground truth, shown below:

> *"Chris Wedge, regista de "L'Era Glaciale" e "Rio", porta nelle sale una fantastica storia ambientata nel mondo, a noi ignoto, dei piccoli esseri che vivono nella natura circostante. Sensazionale."*

We can see that the model understood the content of the input document very well and generated a correct summary, but describes another point of view. We can, therefore, conclude that the model is well trained using the proposed dataset, and that it is able to generalize very well.

As a second example to analyze, we have selected the first example on the MLSum-It test set. In this case, we want to compare the summary produced by the mBART model trained on the IlPost dataset with the summary produced by the same mBART model, but trained on the Fanpage dataset. Using mBART trained on Fanpage, we obtain the following summary:

> *"José Ortega Cano, dopo un mese e mezzo ricoverato nell'ospedale Vergine Macarena di Siviglia dopo aver subito un grave incidente stradale (nel quale è morto Carlos Parra, l'autista dell'altro veicolo coinvolto nel sinistro) ha lasciato ieri mattina il centro ospedaliero dove ha dedicato alcune parole ai numerosi mezzi di comunicazione nazionale."*

When we use the same mBART model, but trained on IlPost, we obtain the following summary:

> *"L'ospedale di Siviglia dove è morto il defunto. José Ortega Cano è in ospedale da un mese e mezzo, dopo un incidente stradale in cui è morto Carlos Parra."*

The first difference we can see in these last two generated summaries is the number of words for each summary: the summary generated with the model trained on Fanpage is longer than that generated with the model trained on IlPost, but both are valid summaries for the input example provided. The second summary has a very assertive first sentence and a second sentence that better explains the content of the article, but this is exactly the style we can find for each summary in the IlPost dataset. Instead, the first summary it is more uniform and has a less journalistic style, but it is equally correct. With this last example, we can easily understand that using metrics such as ROUGE is not enough to evaluate summarization models, because even very different summaries can correctly represent the same input text; thus, a human evaluation is useful.

As the last example to analyze, we show the summary of the second example extracted from the Fanpage test set and generated by the IT5 model trained on Fanpage:

> *"Ll Pentagono ha appena annunciato che sta testando un sofisticatissima IA che ha l'obiettivo di prevedere con "giorni di anticipo" eventi di grande rilevanza nazionale e internazionale, analizzando dati da molteplici fonti."*

The latter is also a good summary, but it contains some grammatical errors such as "un sofisticassima" instad of "una sofisticatissima". Sometimes grammatical errors of this type are generated, so the model is not perfectly trained; however, considering that humans can also make grammatical errors, we can conclude that, in general, good results can be achieved using the proposed datasets.

## 6. Conclusions

In the literature, there are very few papers that address the problem of abstractive text summarization for the Italian language. Here, we propose two new Italian datasets for abstractive text summarization and test them with very recent and powerful deep models.

With this paper, we have tried to answer some questions related to the need to create datasets for the Italian language for text summarization. From the experiments conducted, we can draw the general conclusion that it is always better to create a dataset in the reference language if we want to obtain more robust models. In the event that the creation of a dataset for a particular language becomes a difficult task, then, in this case, it is preferable to automatically translate an existing dataset to the use of a pre-trained model in another language, with a relative translation of its inputs and outputs. Another conclusion that we can draw from this work is that the length of the summaries produced by any model will depend on the length of the summaries available in the training dataset. For this reason, the two datasets proposed here become very interesting for future research activities involving other models for text summarization.

This is only the first step to approach the summarization task in Italian; the proposed dataset covers only some styles of summaries. In the future, we can extend this work by creating other datasets with longer inputs and longer summaries that will allow us to analyze other types of models capable of exploiting all types of input document lengths for summarization tasks.

**Author Contributions:** Conceptualization, N.L., E.F. and I.G.; software, E.F. and N.L.; data curation, E.F.; writing—review and editing, I.G., N.L., E.F. and R.L.G.; supervision, I.G. All authors have read and agreed to the published version of the manuscript.

**Funding:** This research received no external funding.

**Institutional Review Board Statement:** Not applicable.

**Informed Consent Statement:** Not applicable.

**Data Availability Statement:** The datasets used in this paper are in the public domain and can be found in the following links: Fanpage (accessed on 14 October 2021), https://huggingface.co/datasets/ARTeLab/fanpage, IlPost (accessed on 13 October 2021), https://huggingface.co/datasets/ARTeLab/ilpost, MLSum-It (accessed on 14 October 2021), https://huggingface.co/datasets/ARTeLab/mlsum-it (accessed on 14 October 2021).

**Conflicts of Interest:** The authors declare no conflict of interest.

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
