# Peer review of "Two New Datasets for Italian-Language Abstractive Text Summarization"

_information, doi:10.3390/info13050228_

Round 1

Reviewer 1 Report

p.4 eqn 1 is not aligned well.

l.138 corpus -> corpora.

l.145-146 What about longer summaries? Less than 2 sentences is still very short.

p.5 Table 1: is it the number of words or the number of normalized words (terms)?

l.155 Which preprocessing steps? Please describe it in detail.

l.208 goodness -> quality?

l.254: What is 'slightly'? Is there statistical significance?

l.276 and on - these sections are not filled properly.

Author Response

Thanks for the comments and suggestions that lead to an improvement of our paper. 
We have edited the paper following your comments:

p.4 eqn 1 is not aligned well.
    we corrected it.

l.138 corpus -> corpora.
    we corrected it.

l.145-146 What about longer summaries? Less than 2 sentences is still very short.
    The need to have short summaries is very present on the web, both for newspaper articles and for blogs, so we have created datasets following this need. Furthermore, even the data taken into consideration are not particularly long so the summaries are well balanced with respect to the input.
It would be interesting for future work to create datasets with longer input texts and summaries. We added this comment in the conclusions.

p.5 Table 1: is it the number of words or the number of normalized words (terms)?
    we compute the raw number of words.

l.155 Which preprocessing steps? Please describe it in detail.
    we carried out the standard preprocessing implemented in IT5 and mBART, which we do not describe in the paper because we have released the code used to conduct experiments, so it is possible to deepen all the details of this type.

l.208 goodness -> quality?
    we corrected it.

l.254: What is 'slightly'? Is there statistical significance?
    we add numerical specification of that sentence into the paper.

l.276 and on - these sections are not filled properly.
    we fill that sections.

Reviewer 2 Report

In this work, the authors collected data using web-scraping on the sites of news to develop two novel abstractive text summarization datasets for Italian. To compare the proposed datasets, the authors also translated the MLSum dataset to Italian, named MLSum-it. Then several pre-trained deep learning models are used to evaluate the performance on the datasets. The motivation of work is clear, and the proposed datasets are useful to develop and evaluate the text summarization systems in Italian.

Major comments:

  1. The authors collected data via web-scraping on the news site in Italy. The method can be used to other similar sites of news to collect more data. Why only the Fanpage and IlPost are selected? The authors should explain this.
  2. From the experimental results, we can see that the performance of the models trained on the cross dataset is obviously dropped than the one trained on the corresponding training dataset. For example, the ROUGE-1 score of mBart model drops from 38.91 to 29.36 on the IlPost test set. If the two training sets are combined to train the model, will the performance be improved? In other words, does the model generalize better if the model is trained using the combined datasets? The authors should conduct the experiment, i.e., the author can combine the two training datasets to train the model, then evaluate it on the test sets. In addition, will the performance be further improved if the translated dataset is added into training sets?
  3. Besides the evaluation metrics, the authors should show some summary cases for comparison Intuitively. Some error analysis and future works need to be discussed.

Minor comments:

  1. Page 1, line 33, the references of the models should be cited.
  2. Page 5, line 179, “(see list of topics in Table xxx)” should be “(see list of topics in Table 2)”
  3. The proposed datasets are released? The URL should be provided.

Author Response

Thanks for the comments and suggestions that lead to an improvement of our paper. 
We have edited the paper following your comments:

Major comments:

1. The authors collected data via web-scraping on the news site in Italy. The method can be used to other similar sites of news to collect more data. Why only the Fanpage and IlPost are selected? The authors should explain this.

    We chose those websites because they are accredited news sites, so they contain grammatically correct and checked data, which other sites fail to guarantee on all of their posts. Having to download many articles to create the datasets, we cannot afford to have to check all the articles downloaded, so for this reason we have only used reliable and freely available news websites.

2. From the experimental results, we can see that the performance of the models trained on the cross dataset is obviously dropped than the one trained on the corresponding training dataset. For example, the ROUGE-1 score of mBart model drops from 38.91 to 29.36 on the IlPost test set. If the two training sets are combined to train the model, will the performance be improved? In other words, does the model generalize better if the model is trained using the combined datasets? The authors should conduct the experiment, i.e., the author can combine the two training datasets to train the model, then evaluate it on the test sets. In addition, will the performance be further improved if the translated dataset is added into training sets?

    The goal of that experiment is to verify the generalization of the trained model on the datasets created and tested on very different data, certainly by combining the two datasets (which however have different summary modes, which you may have an interest in preserving) there can be improvements, but this would not be useful for demonstrating the effectiveness of the proposed datasets.

3. Besides the evaluation metrics, the authors should show some summary cases for comparison Intuitively. Some error analysis and future works need to be discussed.

    Thanks for your suggestion, we provide some summary examples in the new section "5.1 Human Evaluation" and we discuss them. Also, some future works are commented into the "Conclusion" section.

Minor comments:

    Page 1, line 33, the references of the models should be cited.
        We added the references.
    Page 5, line 179, “(see list of topics in Table xxx)” should be “(see list of topics in Table 2)”
        We corrected the reference.
    The proposed datasets are released? The URL should be provided.
        We added the footnote 4.

Round 2

Reviewer 2 Report

The authors have addressed most of my concerns in this revision. I have no more comments.